# Pulse oximetry adoption and oxygen orders at paediatric admission over 7 years in Kenya: a multihospital retrospective cohort study

Timothy Tuti ,[1] Jalemba Aluvaala ,[1,2] Samuel Akech,[1] Ambrose Agweyu,[1] Grace Irimu,[1,2] Mike English ,[1,3] The Clinical Information Network Author Group

[1]Health Services Unit, KEMRI-Wellcome Trust Research Programme, Nairobi, Kenya
[2]Department of Paediatrics and Child Health, University of Nairobi, Nairobi, Kenya
[3]Nuffield Department of Medicine and Department of Paediatrics, University of Oxford, Oxford, UK

**Correspondence to**
Dr Timothy Tuti;
TTuti@kemri-wellcome.org

## ABSTRACT

**Objectives** To characterise adoption and explore specific clinical and patient factors that might influence pulse oximetry and oxygen use in low-income and middle-income countries (LMICs) over time; to highlight useful considerations for entities working on programmes to improve access to pulse oximetry and oxygen.

**Design** A multihospital retrospective cohort study.

**Settings** All admissions (n=132 737) to paediatric wards of 18 purposely selected public hospitals in Kenya that joined a Clinical Information Network (CIN) between March 2014 and December 2020.

**Outcomes** Pulse oximetry use and oxygen prescription on admission; we performed growth-curve modelling to investigate the association of patient factors with study outcomes over time while adjusting for hospital factors.

**Results** Overall, pulse oximetry was used in 48.8% (64 722/132 737) of all admission cases. Use rose on average with each month of participation in the CIN (OR: 1.11, 95% CI 1.05 to 1.18) but patterns of adoption were highly variable across hospitals suggesting important factors at hospital level influence use of pulse oximetry. Of those with pulse oximetry measurement, 7% (4510/64 722) had hypoxaemia (SpO$_2$ <90%). Across the same period, 8.6% (11 428/132 737) had oxygen prescribed but in 87%, pulse oximetry was either not done or the hypoxaemia threshold (SpO$_2$ <90%) was not met. Lower chest-wall indrawing and other respiratory symptoms were associated with pulse oximetry use at admission and were also associated with oxygen prescription in the absence of pulse oximetry or hypoxaemia.

**Conclusion** The adoption of pulse oximetry recommended in international guidelines for assessing children with severe illness has been slow and erratic, reflecting system and organisational weaknesses. Most oxygen orders at admission seem driven by clinical and situational factors other than the presence of hypoxaemia. Programmes aiming to implement pulse oximetry and oxygen systems will likely need a long-term vision to promote adoption, guideline development and adherence and continuously examine impact.

## INTRODUCTION
### Background

Compared with clinical signs (eg, central cyanosis, indrawing), pulse oximetry is felt to be easy to use and posited to be at least 20%

---

### Strengths and limitations of this study

► The data presented are a rare example of long-term tracking of hospital practice in a routine low-income and middle-income country (LMIC) setting presenting a useful illustration of the process of pulse oximetry adoption in LMIC hospitals that has broader implications for thinking on technology adoption.

► We employ a complex modelling framework to explore patient-level and hospital-level factors influencing outcomes of interest at admission (pulse oximetry done, and oxygen prescribed) over time that accommodate and account for between hospital variability, within hospital non-linear trends in the outcomes over time and entry into the clinical network at different time points.

► We have no contemporaneous data from Kenyan settings that do not belong to the Clinical Information Network for comparison with most other studies exploring the use of pulse oximetry essentially being cross sectional.

---

more accurate in identifying children at risk of hypoxaemia (defined as a blood oxygen saturation of <90%).[1 2] As pulse oximetry is an inexpensive tool, it is now recommended for guiding the assessment of illness severity and treatment for respiratory illnesses, including increasing emphasis on its role for screening children with respiratory illness in primary care in low-income and middle-income country (LMIC).[1 2] As hypoxaemia may be found in all severe childhood illnesses such as sepsis, meningitis, severe acute malnutrition and malaria, its use is widely recommended as a screening tool in all paediatric admissions as a fifth vital sign. The advent of COVID-19 has further invigorated programmatic efforts to scale up pulse oximetry together with improved oxygen delivery systems.[3–9]

Despite being recommended for many years, there is little long-term data on the adoption of pulse oximetry in LMIC, even

from hospitals. Data on use during the COVID-19 pandemic are also lacking from such settings as are large studies from routine settings on whether pulse oximeter readings influence the use of oxygen. We and others have described that adoption is undermined by system factors such as inadequate supply and repair of oximeters and if healthcare workers have insufficient training on when, how, and why to use them and interpret their results.[9 10] This compounds wider and similar problems in managing oxygen systems themselves.[11] Our earlier work used data from seven Kenyan hospitals between September 2013 and February 2016. Here, we update and extend these analyses to include data from paediatric admissions in a growing network of Kenyan hospitals before and after prolonged national health workers strikes in 2017, including data obtained during the COVID-19 pandemic.[12] The objective of this study was to characterise adoption and explore specific patient, clinical and hospital factors that might influence pulse oximetry and oxygen use and highlight useful considerations for entities working on programmes to improve access to pulse oximetry and oxygen.[13–15]

## METHODS
### Ethics and reporting
The reporting of this observational study follows the Strengthening the Reporting of Observational Studies in Epidemiology statement.[16] The Scientific and Ethics Review Unit of the Kenya Medical Research Institute (KEMRI) approved the collection of the deidentified data that provides the basis for this study as part of the Clinical Information Network (CIN). The CIN is run in partnership with the Ministry of Health and participating hospitals with aims to improve the quality of routine paediatric hospital data for use in improvement activities, observational and interventional research.[17 18] Individual consent for access to deidentified patient data was not required.

### Patient and public involvement
No patients were involved in the design, conduct, reporting or dissemination plans of our research except through the KEMRI ethical review process where they have representatives.

### Study design and setting
From a broad context perspective, in many LMICs including Kenya, hospital management and monitoring systems are weak, with major human and material resource constraints. These challenges affect hospitals' delivery of inpatient maternal, surgical and adult medical care as well as paediatric and neonatal care. Consequently, there is very limited organisational and resource slack to mobilise for any new purpose. Interventions such as oximetry and oxygen seeking to achieve large scale change must therefore either consider how to mobilise new resources or consider what is achievable with limited resources.[19 20] We have described the broader context of the Kenyan healthcare system with reference to the paediatric burden of disease in great detail elsewhere.[19 20]

We report a retrospective cohort study of 18 public hospitals in Kenya largely providing first-referral level care predominantly admitting patients that present directly to the facility, and purposefully selected to be of at least moderate size and representative of different malaria transmission zones. Hospitals joined the CIN at different calendar time points between 2013 and 2017. Few of their patients may be formally referred from primacy care facilities but there are few functional referral mechanisms such as ambulance systems.[21] Pulse oximetry has generally not been available outside hospitals in the public sector.[22] Pneumonia is the major killer of children across the country except in settings where malaria is highly endemic (9/18 hospitals in this study).[21]

The hospitals receive 3 monthly clinical audit and feedback reports on the quality of care they provide for common conditions.[23] Paediatric team leaders (paediatricians and nurses) met face to face once or twice annually until 2019 (before the pandemic) to discuss these reports and how to improve clinical care. From 2018, participation in more specific research studies was also discussed in meetings with hospitals. This resulted in two studies being initiated in subsets of the CIN hospitals which might have influenced the adoption of pulse oximetry. Hospital participation in these studies was not mutually exclusive: a hospital could be part of one, both or none of the research studies. Not all hospitals recruited in the same study were recruited at the same time. Summary details of these studies are provided in table 1 with greater detail on which hospitals participated in the studies provided in online supplemental tables 1 and 2. The 18 hospitals included in the study had a median of 1 pulse oximeter(s) per paediatric ward (IQR: 1–3) (online supplemental table 1).

Recording of pulse oximetry values has been included as part of a structured paediatric admission record (PAR)[24] used since 2013. Hospitals joining the CIN agree to provide the PAR themselves and promote its use—the purpose of the PAR is to prompt admitting clinicians to fully assess children and rapidly document their findings.[24] Emergency Triage Assessment and Treatment plus Admission Care (ETAT+) training[25] has been used in Kenya since 2008 and was adapted in 2013 so that pulse oximetry was a recommended part of the assessment in all sick children and especially those with danger signs. Many of the junior and senior medical staff in CIN hospitals would have received ETAT+training (eg, as an undergraduate or postgraduate course or as in-service training).[25] Additionally, the Basic Paediatric Protocols that are widely disseminated have specifically referred to the use of pulse oximetry—if available—for all pneumonia cases since 2016.[26] The promotion of PAR use by hospitals, adaption and scale up of ETAT+training, and dissemination of the basic paediatric protocols to clinicians nationally are posited to have a system-wide effect on the adoption of pulse oximetry use.

**Table 1** Description of studies undertaken in Clinical Information Network hospitals

| Study name | Study site selection rationale and period | Goal | Extra resources provided to hospitals (H) by projects* | Extra notes |
|---|---|---|---|---|
| RTS, S/AS01 trial: (ClinicalTrials.gov Identifier: NCT03806465) | Study start: December 2018 Site selection: by MoH/WHO, selecting high volume public hospitals in regions getting the vaccine. | The evaluation of the feasibility, safety and impact of the RTS, S/AS01 malaria vaccine in paediatric cases in Kenya | One extra healthcare worker provided per paediatric ward. Extra pulse oximetry equipment provided per ward: ▶ One item: H4 | Clinician role: promote compliance with guidelines for meningitis and malaria including complete assessment and investigations including the performance of lumbar punctures, and consenting participants for storage of cerebrospinal fluid storage |
| SEARCH trial (ClinicalTrials.gov Identifier: NCT04041791) | Started in two phases: ▶ March 2019 Hospitals: H3, H9, H12, H13, H16 ▶ June 2019 Hospitals: H1, H2, H6, H10, H11 Site selection: from the expected size of pneumonia population, the geographical balance of hospitals and support from local hospital teams | Pragmatic factorial individually randomised controlled trial to compare alternative antibiotics and alternative modes of fluid therapy for children with severe pneumonia | One extra healthcare worker provided per participating hospital. Extra pulse oximetry equipment provided to the hospital: ▶ Two items: H3, H6, H9, H10, H11, H12, H13, H16 ▶ One item: H1, H2 | Clinician role: promote compliance with the study protocol, including completing patient assessment, investigations and diagnoses. Ensure appropriate administration of study interventions. Recruitment of study participants was suspended on the 8 April 2020 due to COVID-19 pandemic. Adult inpatient surveillance in addition to paediatric surveillance began on the 4 May 2020. |

H15 got one pulse oximeter in June 2020 even though it was not part of any study.
*We were not able to track and record introduction of pulse oximeters procured directly by hospitals or other sources, or those personally owned by healthcare workers, nor ability to repair existing pulse oximeters when required.
MoH, Ministry of Health.

## Study participants

All medical admissions to paediatric wards (ages 0–13 years), but not those admitted to specific newborn units, to the selected hospitals between 1 February 2014, or the date at which a hospital joined CIN, and 31 December 2020 were eligible for inclusion in this study. We excluded children whose admission or discharge dates were missing or improbable (eg, discharge date is earlier than admission date), and those whose admission fell within 2017, a period of prolonged health worker strikes that resulted in major disruption to healthcare delivery.[12] The expectation was that all would have pulse oximetry performed at admission given they were sick enough to warrant inpatient care but among these we also report the proportion of children with danger signs comprising any level of altered consciousness or with recorded signs of respiratory distress especially deserving of pulse oximetry.[10 26 27]

## Data sources and management

Methods of collection and cleaning of data in the CIN are reported in detail elsewhere.[28] Clinical data for paediatric admissions to the hospitals within the CIN are captured through PAR forms[27] that are approved by the Ministry of Health. The PAR prompts the clinician with a checklist of fields including patient biodata, clinical assessment, admission and discharge diagnoses, treatments and to record outcome (survival or death). The CIN supports one data clerk in each hospital to extract data from paper medical records, nursing charts, treatment charts and available laboratory reports each day after children's discharge into the primary data collection tool developed in Research Electronic Data Capture. Automated error checking happens at the point of entry by daily review, every week centrally and both are complemented by regular external data quality assurance reviews.[28] A minimal dataset—which is unsuitable for these pulse oximetry analyses—is collected for (1) all admissions with a burn or a surgical diagnosis to the paediatric ward(s), (2) admissions during major holiday breaks, (3) admissions when the data clerk was on leave and (4) on a random selection of records in hospitals where the workload is very high. This process is explained in detail elsewhere.[28 29]

## Descriptive analysis

To characterise adoption of pulse oximetry at admission over time, we use pooled data from the period February 2014 to November 2018 (excluding 2017) from 13 network hospitals. From December 2018 data were available from 17 hospitals after 5 joined the CIN linked to new studies, and one (H7) exited. To explore patterns in more detail, we employed two approaches: (1) for the period from December 2018 to December 2020, we plot data pooled for hospital subgroups depending on which research studies they were taking part in as these might influence pulse oximetry use; (2) we plot adoption within each hospital for the number of months it was part of the network, again excluding 2017 where relevant. This latter approach was used to reveal the extent of variability in adoption at the hospital level.

For the individual patient population, we tabulated and summarised categorical data as proportions, with continuous variables reported as medians (IQR) if they were not normally distributed. From the included paediatric admission patient population data, we report the prevalence of important clinical features such as respiratory

distress symptoms (eg, central cyanosis, indrawing, grunting and difficulty breathing), circulatory symptoms (eg, slow capillary refill, less than three seconds), level of consciousness, and whether the patient was being readmitted or referred to the hospital. We also report the prevalence of summary clinical features such as having any sign of respiratory distress and any danger sign and the proportion for whom oxygen was ordered as a treatment at admission.

### Statistical methods

To examine two binary outcomes, (1) pulse oximetry used and, (2) oxygen ordered at admission, we modelled individual patient data. Our key explanatory variable was time as part of (or exposure to) the CIN, computed as the number of months since joining the CIN for a specific hospital. This allowed associations with these outcomes to be explored in hospitals while accounting for clustering by both hospital and time in months since the hospital joined the CIN; the analysis of pulse oximetry use employs a much larger dataset to extend our earlier findings.[10]

Predictors included in our models were patients' age and sex, and the record of presence or absence of respiratory signs and symptoms, circulatory symptoms, level of consciousness, whether the patient was being readmitted or referred to the hospital. At the hospital level, we included terms for their specific identity and malaria endemicity zone. Hospital participation in one of the four research study subgroups was included as a covariable at the patient level to take account of possible effects occurring only after a hospital joined specific studies.

We modelled whether oxygen was prescribed at admission with an interest in whether patients with a recorded pulse oximetry value ($SpO_2$ <90%) at admission would be started on oxygen therapy. Where the pulse oximetry value at admission was missing, based on findings from a previous CIN study, the assumption was that the clinician did not have the information.[10] The analysis therefore explored (1) whether pulse oximetry values ($SpO_2$ <90%) were associated with oxygen prescription and (2) in the absence of a pulse oximetry value <90%, which clinical signs are associated with the prescription of oxygen therapy.

We included the variables of interest in hierarchical multivariate logistic regression models, with a random intercept for each hospital and a random slope for the time within hospital. This model specification commonly referred to as a growth curve model, allowed the explanatory variable of time to have a different effect for each hospital. A *growth curve model* typically refers to statistical methods that allow, in our case, the estimation of interhospital variability in intrahospital patterns of change over time.[30] Our approach to the growth-curve model fitting is within the multilevel modelling framework, with patients nested in hospitals nested in time points.[30] Different ways of specifying the growth-curve model using a multilevel modelling framework are explained in detail elsewhere.[31] Before embarking on the growth-curve modelling, we examined for each hospital whether the 2017 strike resulted in a significant discontinuity in pulse oximetry use to check if treating time as months in the CIN, a continuous variable, instead of as calendar time was justifiable given the strike caused a 12-month data gap.

Where there were missing patient-level data, we applied fully conditional specification multivariate imputation by chained equations[32] to generate imputed datasets under the missing at random (MAR) assumption to allow us to analyse all eligible patients. Previous studies indicate that MAR is a reasonable approach for CIN data.[33] We contrasted findings using imputation with the findings of the complete case analysis.

## RESULTS

### Descriptive findings

Figure 1 depicts the study population inclusion process. Out of the 179 991 paediatric admissions to CIN hospitals, 132 737 (73.7%) were eligible for analysis. Most exclusions were because an admission was randomly sampled for minimum data collection (n=33 074, 70% of the exclusions) or fell in the strike year (n=8787, 19% exclusions). Of the population included in the analysis, 113 196/132 737 (86.1%) demonstrated at least one sign indicating increased risk of hypoxaemia (respiratory distress symptoms or altered consciousness, or age below 1 month). Given this high proportion, we report results based on data from all eligible admissions (figure 1).

Of the 132 737 cases, 48.8% (n=64 722) had a pulse oximetry measurement taken at admission. Of the 64 722 patients with pulse oximetry measured at admission, 4510 (7%) had hypoxaemia ($SpO_2$ <90%). Across the same period, 11 428/132 737 (8.6%) were documented to have oxygen prescribed at admission. Of these, only 1484/11 428 (12.8%) were confirmed to be hypoxaemic using pulse oximetry, while 6633/11 428 (58%) had no pulse oximetry measurement and 3311/11 428 (28.9%) were documented to have a pulse oximetry value of 90% or higher at admission. Thus, 87% oxygen use was not pulse oximetry guided ($SpO_2$ ≥90% or unknown) and more than twice as many children prescribed oxygen had values ≥90% (3311 compared with 1484) (figure 2).

From February 2014 to December 2016, on average, there was a steady improvement in pulse oximetry adoption at admission in the 13 hospitals from between 10% and 20% in 2014 to between 50% and 70% at the end of 2016 (figure 3), even though no pulse oximetry devices were provided through CIN. Following the strikes in 2017, from January 2018 when services resumed up until November 2018, average pulse oximetry adoption in these 13 sites seemed to stabilise at around 50%. By December 2018, one hospital exited and five hospitals joined CIN and different research studies were initiated in subsets of hospitals (table 1, figure 3 and online supplemental table 2). Pulse oximetry adoption ranged from an average of 40% to almost 100% across these subsets (figure 3).

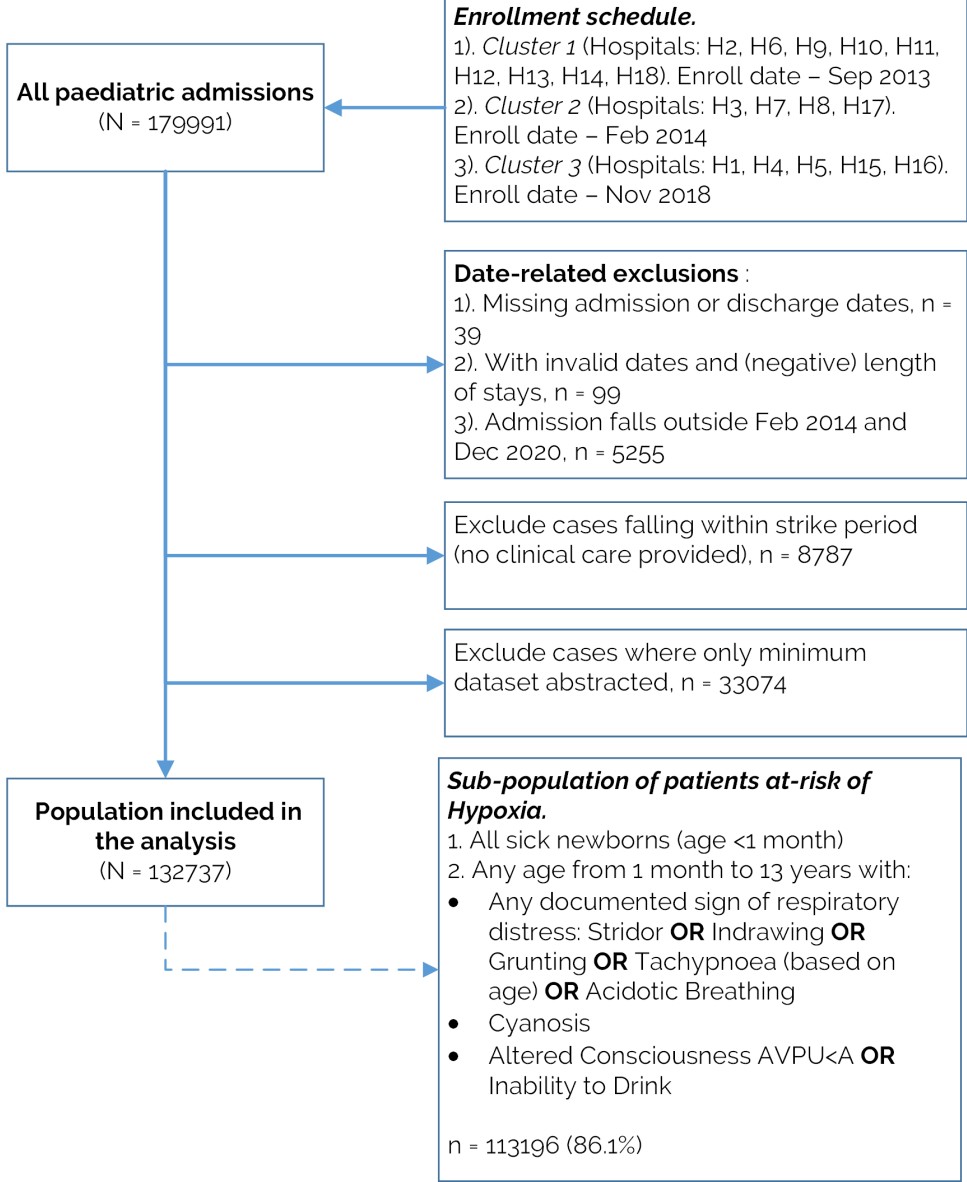

**Enrollment schedule.**
1). *Cluster 1* (Hospitals: H2, H6, H9, H10, H11, H12, H13, H14, H18). Enroll date – Sep 2013
2). *Cluster 2* (Hospitals: H3, H7, H8, H17). Enroll date – Feb 2014
3). *Cluster 3* (Hospitals: H1, H4, H5, H15, H16). Enroll date – Nov 2018

**All paediatric admissions**
(N = 179991)

**Date-related exclusions** :
1). Missing admission or discharge dates, n = 39
2). With invalid dates and (negative) length of stays, n = 99
3). Admission falls outside Feb 2014 and Dec 2020, n = 5255

Exclude cases falling within strike period (no clinical care provided), n = 8787

Exclude cases where only minimum dataset abstracted, n = 33074

**Population included in the analysis**
(N = 132737)

**Sub-population of patients at-risk of Hypoxia.**
1. All sick newborns (age <1 month)
2. Any age from 1 month to 13 years with:
- Any documented sign of respiratory distress: Stridor **OR** Indrawing **OR** Grunting **OR** Tachypnoea (based on age) **OR** Acidotic Breathing
- Cyanosis
- Altered Consciousness AVPU<A **OR** Inability to Drink

n = 113196 (86.1%)

**Figure 1** Flow chart of the inclusion criteria. AVPU, Alert, Verbal, Pain, Unresponsive.

Exploring each hospital's adoption trajectory illustrates in many a general improvement in adoption throughout their participation in CIN but also examples of adoption failure (figure 4). Figure 4 tends to confirm as depicted in figure 3, that despite a strike of almost 1 year there was minimal evidence of discontinuity in hospitals' pattern of adoption from 2016 to 2018. The lack of evidence for discontinuity (supported by additional analyses in online supplemental table 3) supports our treatment of time as 'months in CIN' as a continuous variable in place of calendar time in figures 3 and 4 and subsequent models.

From figure 4, 8 of the 13 hospitals recruited into the CIN by February 2014 had no use of pulse oximetry at admission at all (median at baseline 0%), with the highest baseline use being 65% (H13). For the hospitals joining after the strike (ie, from November 2018), use of pulse oximetry during admission was generally low for 4/5 hospitals (median at baseline 25%) but one hospital already used pulse oximetry on 83% admissions (H16) (figure 4). Hospitals demonstrated various patterns of adoption: (1) low adoption when joining the CIN with a sharp improvement later (eg, hospital H2, H12), (2) oscillation between low and high pulse oximetry use (eg, hospital H3, H10, H17), (3) gradual improvement that later stabilised (eg, hospital H1, H4, H14) and (4) failure of adoption (H7 and H5) (figure 4). These varying patterns of adoption, especially after December 2018, do not clearly seem to be related to study participation. Hospitals participating in both new studies, resulting in presence of two additional non-physician clinician research team members, did demonstrate a consistent improvement in pulse oximetry use at admission (figures 3 and 4). However, pulse oximetry adoption at admission was also high in hospitals not participating in any study within the CIN (figures 3 and 4).

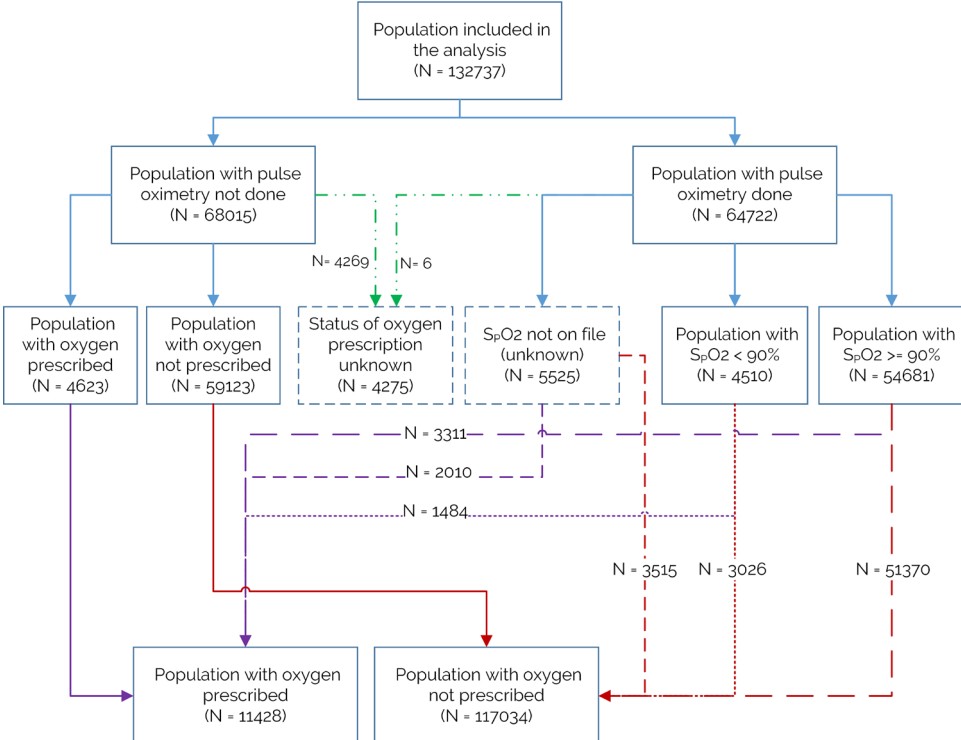

**Figure 2** Oxygen prescription and use of pulse oximetry at admission.

### Individual patient data and exploratory modelling

Table 2 provides the characteristics of the children admitted to these hospitals, including the level of missingness in each variable. From the study population, 32% had at least one missing variable, with the median number of variables missing per patient of 2 of 30 (IQR: 0–3). None of the individual variables was missing more than 30% of values. In this paediatric admission population, the prevalence of chest indrawing, tachypnoea and grunting is 22.4%, 33% and 10.3%, respectively, and inability to drink (or breastfeed), a danger sign requiring urgent attention, was present in 72.7% of the admissions; 82.9% of all the admissions had at least one of these signs. The median length of stay was 3 days (IQR: 2–6), with a median of one additional comorbid condition at admission (IQR: 0–1). Pneumonia and malaria were the leading, often comorbid, admission diagnoses (online supplemental figure 1).

Stridor, wheezing, acidotic breathing and cyanosis have relatively low recorded prevalence of 2%, 5.3%, 2.3% and 0.7%, respectively and were not included in our previous models.[10] Whether a child is a readmission or referral from another health facility are also newly added data with higher prevalence at 10.2% and 14.7% of the patient population, respectively[10] (table 2).

### Summary of the growth curve model results

Table 3 provides the results of the growth curve models, where adoption rates between hospitals were adjusted for malaria endemicity, participation in research studies, participants' demographic characteristics and clinical signs and symptoms at admission. These results are from

analysis using 35 imputed datasets (32% of the cohort had at least one missing variable).

Associations of the hospital and clinical features with the outcomes from table 3 are time-invariant (ie, independent of the effect of time in months in CIN), except for the interaction term of PAR use with time. From online supplemental table 4 and figure 4, the models' high intraclass correlation coefficient (ICC) is indicative that hospital factors play a bigger role in the adoption of pulse oximetry over time compared with individual patient-level factors. The contribution of hospital level factors is less pronounced for oxygen prescription at admission. The relatively high ICC is not unexpected in longitudinal models with measurements clustered by hospital and given the varying patterns of adoption seen in figure 4.

### Association with the use of pulse oximetry at admission

Over the entire study period and across all hospitals pulse oximetry use at admission improved as time participating in the CIN increased (month-to-month improvement OR: 1.11, 95% CI 1.05 to 1.18). This was likely driven by noticeable improvement in 10 hospitals and in 5 of these specifically there were substantive positive changes in adoption of pulse oximetry on a month-to-month basis; (online supplemental figure 2). Those improving most tended to have very low use of oximetry at the time they joined the CIN (online supplemental figure 2). This improvement was seen in tandem with the improvement in PAR use which might be expected to encourage better assessment and potentially, therefore, aid clinical decision-making.

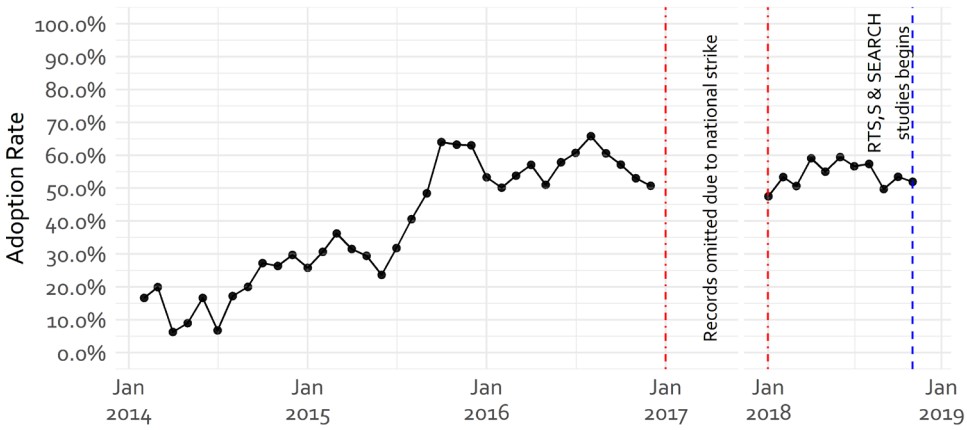

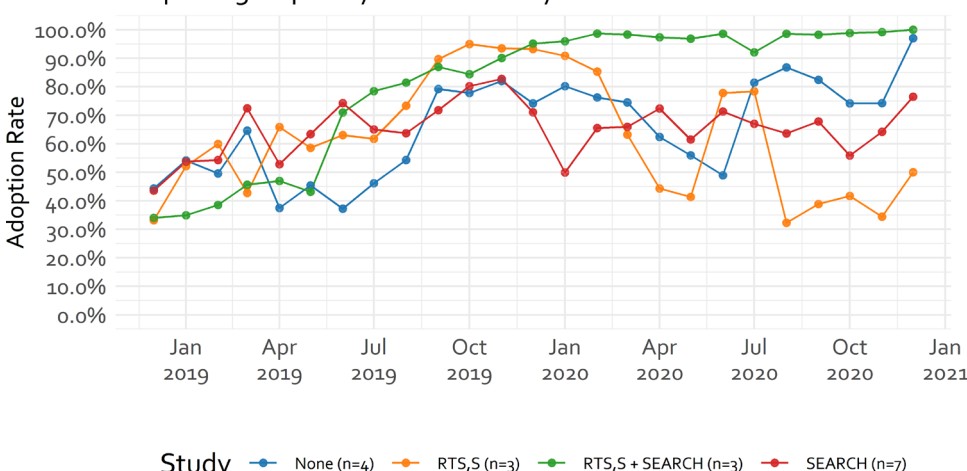

**Figure 3** Average pulse oximetry use in the Clinical Information Network (CIN) hospitals over time. The number of hospitals in each study covers both presently involved and those joining the CIN later: there are 13 hospitals in the period between February 2014 and November 2018, with the lower panel representing transition to 18 hospitals by adding 5 hospitals and research studies beginning. The 5 new hospitals' distribution among the studies in the following way: RTS,S: 2/5; RTS,S+SEARCH: 1/5; SEARCH: 1/5; None: 1/5.

From table 3, patients admitted with signs of fever, cough, difficulty breathing, chest indrawing, crackles and inability to drink had odds statistically greater than one of having pulse oximetry measurement taken at admission over time than those without these signs. With a larger dataset, the inclusion of new variables and adjusting for the time in the CIN, our findings conflict with previous findings on the significance of tachypnoea, altered consciousness (AVPU <A), and inability to drink as signs associated with the use of pulse oximetry at admission with a reversal in association (eg, tachypnoea, inability to drink) or no statistically appreciable association (eg, AVPU <A).[10] Grunting and acidotic breathing symptoms, likely to be associated with malaria and anaemia (online supplemental figure 1), were statistically, not significantly

associated with pulse oximetry use at admission (table 3). Cyanosis, grunting and acidotic breathing—which were previously not reported[10]—were significantly associated with the odds of pulse oximetry done at admission being lower than one. Our findings also highlight that referral into the hospital is positively associated with the use of pulse oximeters at admission (OR: 1.05, 95% CI 1.01 to 1.09) (table 3) and that patients admitted to hospitals that were part of an active research study were 49% more likely to have a pulse oximetry measurement taken at admission than in hospitals not participating in any study (table 3). Additional analyses suggest the more clinical signs a child has of respiratory distress the more likely they were to be hypoxaemic (online supplemental figure 3).

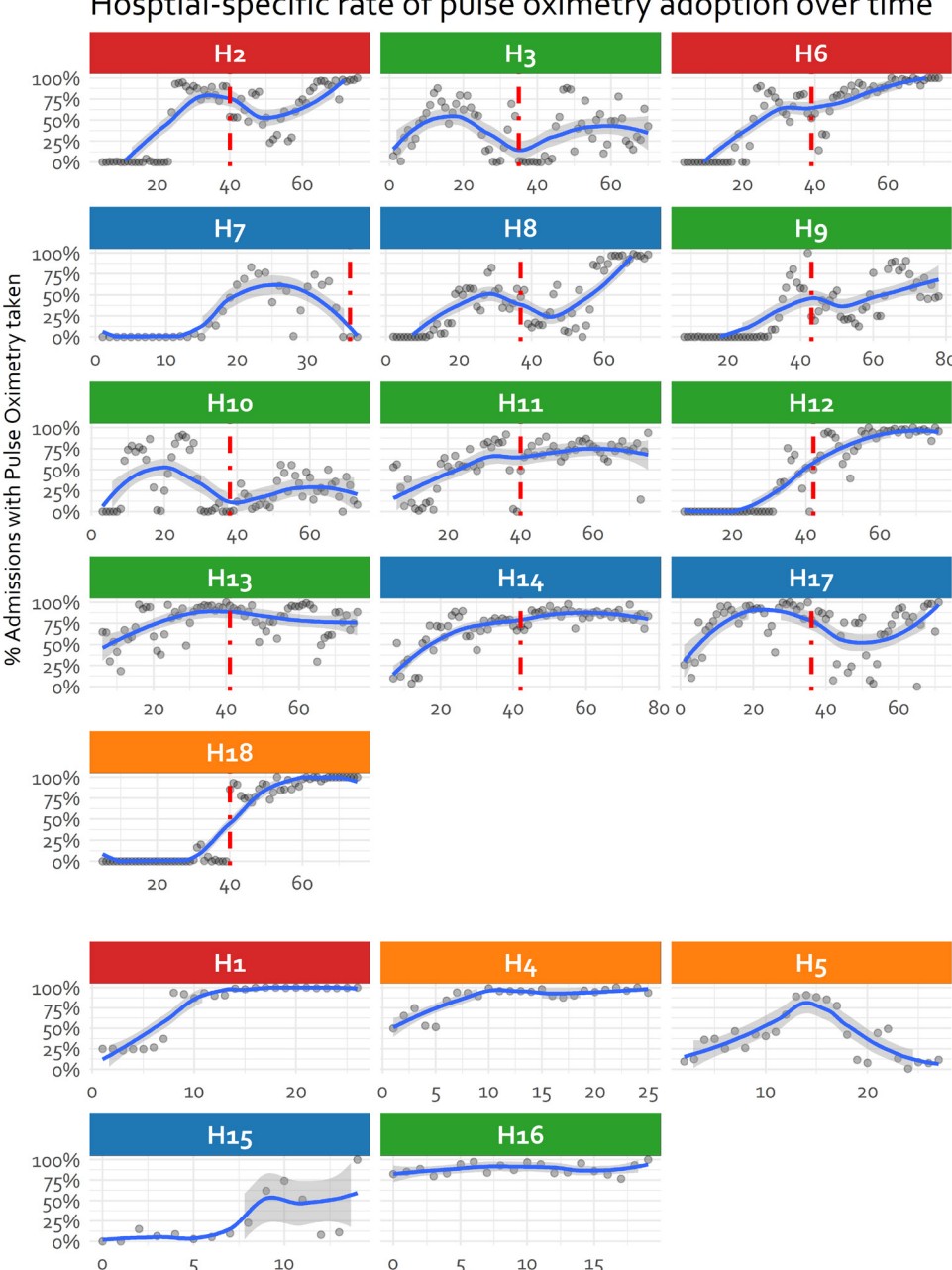

## Hosptial-specific rate of pulse oximetry adoption over time

**Figure 4** Hospital specific rate of pulse oximetry use at admission over time divided into sites joining from January 2014 and those recruited from December 2018. Vertical red line indicates national strike disruption. The title is colour-coded to reflect the research study the hospital was recruited into (Blue: None, Orange: RTS,S, Red: SEARCH, Green: RTS,S+SEARCH). H7 exited from Clinical Information Network and is therefore not included in subsequent analysis.

### Associations with oxygen prescription at admission

Admission with any sign of respiratory distress where the pulse oximetry measure is unknown is associated with oxygen prescription at admission (table 3). The specific signs that appear to prompt oxygen prescription, in reducing magnitude of association, include difficulty breathing, indrawing, AVPU <A, grunting and tachypnoea. Admissions with altered consciousness (AVPU <A) had almost twice the odds of oxygen prescription at admission, independent of respiratory signs (table 3). Interestingly, the direction of association with oxygen

prescription is inverse to that for use of pulse oximetry for: age of the child, signs of cyanosis, grunting, acidotic breathing, AVPU <A, pallor, tachypnoea, fever and inability to drink.

With the inclusion of new variables and adjusting for the time in CIN, our findings affirm previous analyses but show the magnitude of association of cough and difficulty breathing with oxygen prescription at admission to be higher than previously reported.[10] Acidotic breathing, stridor and wheezing, clinical signs omitted from the previous study, were shown to be statistically significant

**Table 2** Descriptive summary statistics for the variables of the patient population included in the study

| Indicator | Levels | Number, n (%) | Missing data, n (%) |
|---|---|---|---|
| **Hospital-level factors** | | | |
| Malaria endemicity zone | High | 53 177 (40.1%) | |
| | Low | 79 560 (59.9%) | |
| **Patient-level factors** | | | |
| Age* | <1 month | 3321 (2.5%) | 1221 (0.9%) |
| | 1–11 months | 38 235 (28.8%) | |
| | 12–59 months | 64 628 (48.7%) | |
| | ≥5 years | 25 332 (19.1%) | |
| Female | Present | 58 588 (44.1%) | 1199 (0.9%) |
| Case is readmission | Present | 13 557 (10.2%) | 37 002 (27.9%) |
| Case is referral to hospital | Present | 19 525 (14.7%) | 39 879 (30%) |
| Presence of comorbidity | Present | 82 918 (62.5%) | 725 (0.5%) |
| AVPU†=A | Present | 113 378 (85.4%) | 10 664 (8%) |
| Slow capillary refill | Present | 2231 (1.7%) | 21 902 (16.5%) |
| Pallor | None | 95 612 (72%) | 10 799 (8.1%) |
| | Mild/moderate | 15 596 (11.7%) | |
| | Severe | 10 730 (8.1%) | |
| Fever | Present | 105 268 (79.3%) | 6952 (5.2%) |
| Cough | Present | 64 819 (48.8%) | 9237 (7%) |
| Crackles | Present | 22 835 (17.2%) | 11 695 (8.8%) |
| Central cyanosis | Present | 974 (0.7%) | 9758 (7.4%) |
| Acidotic breathing | Present | 3098 (2.3%) | 12 740 (9.6%) |
| Difficulty breathing | Present | 38 825 (29.2%) | 10 454 (7.9%) |
| Grunting | Present | 13 701 (10.3%) | 12 255 (9.2%) |
| Tachypnoea‡ | Present | 43 832 (33%) | 31 136 (23.5%) |
| Indrawing | Present | 29 698 (22.4%) | 11 419 (8.6%) |
| Wheezing | Present | 7085 (5.3%) | 11 034 (8.3%) |
| Stridor | Present | 2719 (2%) | 14 121 (10.6%) |
| Inability to drink | Present | 96 455 (72.7%) | 14 846 (11.2%) |
| Weight-for-age Z-score (WAZ) | > −2SD (normal) | 74 035 (55.8%) | 33 486 (25.2%) |
| | −2SD to −3SD (low) | 12 377 (9.3%) | |
| | < −3SD (very low) | 12 839 (9.7%) | |
| Pulse oximetry done at admission‡ | Present | 64 722 (48.8%) | |
| Oxygen therapy prescribed at admission | Present | 11 428 (8.6%) | 4275 (3.2%) |
| Outcome at discharge | Alive | 123 845 (93.3%) | 499 (0.4%) |
| | Died | 8393 (6.3%) | |

*<1 month represents neonates admitted to paediatric ward; no data from the new-born units were included.
†AVPU scale is an acronym for 'Alert, Verbal, Pain, Unresponsive' and is a measure of a patient's level of consciousness ordered by severity from left to right, with 'A' being the least severe.
‡Symptom is derived and calculated as present where either (1) respiratory rate >59 and age <1 month, (2) respiratory rate >49 and age is 1–11 months or (3) respiratory rate is >39 and age is 12 months or older.
§If missing, it was assumed that it was not done. 5525 admission cases reported having done it but did not have the $SpO_2$ value on the paper record.

and positively associated with the odds of being prescribed oxygen at admission. Grunting and acidotic breathing, likely to be associated with malaria and anaemia diagnoses, were not statistically associated with having pulse oximetry measurement taken at admission but were strongly associated with the prescription of oxygen at admission (table 3).

Our findings also highlight that being a patient referred into the hospital (OR: 1.19, 95% CI 1.12 to 1.26) or admitted during the weekend (OR: 1.07, 95% CI 1.02

**Table 3** Growth curve model of pulse oximetry adoption and orders for oxygen use at admission

| Predictors | Pulse oximetry done? | | | Oxygen prescribed? | | |
|---|---|---|---|---|---|---|
| | OR | 95% CI | P value | OR | 95% CI | P value |
| (Intercept)* | 0.03 | 0.01 to 0.17 | <0.001 | 0.01 | 0.01 to 0.02 | <0.001 |
| Hospital factors | | | | | | |
| Malaria endemicity zone: high (ref: low) | 1.62 | 0.1 to 25.04 | 0.729 | 1.55 | 0.9 to 2.67 | 0.112 |
| Time (months in CIN) | 1.11 | 1.05 to 1.18 | <0.001 | 1.02 | 1 to 1.03 | 0.045 |
| Patient-level factors | | | | | | |
| Ref: patient in any study (No)† | | | | | | |
| Patient in any study (prestudy period)‡ | 2.79 | 2.58 to 3.02 | <0.001 | 1.02 | 0.9 to 1.15 | 0.800 |
| Patient in any study (Yes) | 1.46 | 1.35 to 1.58 | <0.001 | 1.02 | 0.91 to 1.15 | 0.715 |
| Referral: Yes (ref: No) | 1.05 | 1.01 to 1.09 | 0.025 | 1.19 | 1.12 to 1.26 | <0.001 |
| Readmission: Yes (ref: No) | 1.01 | 0.97 to 1.05 | 0.732 | 1.07 | 1 to 1.15 | 0.052 |
| Is a weekend admission: Yes (ref: No) | 0.98 | 0.95 to 1.01 | 0.144 | 1.07 | 1.02 to 1.13 | 0.009 |
| PAR used: Yes (ref: No) | 1.99 | 1.79 to 2.21 | <0.001 | 1.23 | 1.02 to 1.48 | 0.031 |
| PAR used * time (months in CIN) | 1.01 | 1 to 1.01 | <0.001 | 1 | 1 to 1.01 | 0.562 |
| Ref: age (>59 months) | | | | | | |
| Age (<1 month) | 0.46 | 0.41 to 0.51 | <0.001 | 1.96 | 1.62 to 2.35 | <0.001 |
| Age (1–11 months) | 1.08 | 1.03 to 1.13 | <0.001 | 1.43 | 1.32 to 1.54 | <0.001 |
| Age (12–59 months) | 1.09 | 1.05 to 1.13 | <0.001 | 1.08 | 1 to 1.16 | 0.054 |
| Female: Yes (ref: No) | 1 | 0.97 to 1.02 | 0.723 | 1.02 | 0.97 to 1.06 | 0.424 |
| Fever: Yes (Ref: No) | 1.18 | 1.14 to 1.23 | <0.001 | 0.78 | 0.73 to 0.83 | <0.001 |
| Cough: Yes (Ref: No) | 1.04 | 1.01 to 1.08 | 0.008 | 1.36 | 1.29 to 1.44 | <0.001 |
| Difficulty breathing: Yes (ref: No) | 1.06 | 1.02 to 1.1 | 0.002 | 2.79 | 2.64 to 2.94 | <0.001 |
| Stridor: Yes (ref: No) | 1.05 | 0.95 to 1.16 | 0.325 | 1.14 | 1.02 to 1.27 | 0.017 |
| Cyanosis: Yes (ref: No) | 0.77 | 0.65 to 0.92 | 0.004 | 1.5 | 1.27 to 1.77 | <0.001 |
| Indrawing: Yes (ref: No) | 1.25 | 1.2 to 1.31 | <0.001 | 2.36 | 2.23 to 2.49 | <0.001 |
| Grunting: Yes (ref: No) | 0.88 | 0.84 to 0.93 | <0.001 | 1.9 | 1.8 to 2 | <0.001 |
| Acidotic breathing: Yes (ref: No) | 0.89 | 0.8 to 0.98 | 0.018 | 1.34 | 1.22 to 1.47 | <0.001 |
| Wheezing: Yes (ref: No) | 1.05 | 0.98 to 1.12 | 0.167 | 1.13 | 1.06 to 1.21 | 0.001 |
| Crackles: Yes (ref: No) | 1.06 | 1.01 to 1.11 | 0.010 | 1.25 | 1.19 to 1.32 | <0.001 |
| Tachypnoea: Yes (ref: No) | 0.96 | 0.93 to 0.99 | 0.022 | 1.59 | 1.51 to 1.68 | <0.001 |
| Alert (AVPU<A): No (ref: Yes) | 0.96 | 0.9 to 1.03 | 0.238 | 1.95 | 1.81 to 2.1 | <0.001 |
| Inability to drink: Yes (ref: No) | 1.04 | 1 to 1.08 | 0.043 | 0.8 | 0.76 to 0.85 | <0.001 |
| Ref: pallor (none) | | | | | | |
| Pallor (mild/moderate) | 0.93 | 0.89 to 0.97 | 0.002 | 1.1 | 1.02 to 1.17 | 0.008 |
| Pallor (severe) | 0.94 | 0.89 to 0.99 | 0.024 | 1.3 | 1.19 to 1.42 | <0.001 |
| Ref: WAZ (normal) | | | | | | |
| WAZ (low) | 0.97 | 0.92 to 1.01 | 0.135 | 0.98 | 0.91 to 1.05 | 0.551 |
| WAZ (very low) | 0.94 | 0.9 to 0.99 | 0.020 | 0.93 | 0.87 to 1 | 0.059 |
| Slow capillary refill: Yes (ref: No) | 0.88 | 0.78 to 1 | 0.051 | 0.97 | 0.83 to 1.13 | 0.693 |
| Ref: hypoxaemia (unknown) | | | | | | |

**Table 3** Continued

| Predictors | Pulse oximetry done? | | | Oxygen prescribed? | | |
|---|---|---|---|---|---|---|
| | OR | 95% CI | P value | OR | 95% CI | P value |
| Hypoxaemia (No) | | | | 0.59 | 0.56 to 0.62 | <0.001 |
| Hypoxaemia (Yes) | | | | 1.92 | 1.77 to 2.08 | <0.001 |

*Represents the average odds that a patient will have pulse oximetry is done (or oxygen therapy is prescribed) at admission in this sample when all the predictors are set to their reference levels (eg, 'No') at the first month of being a member of the Clinical Information Network (CIN).

†'*Patient is in any study*' point estimates and CIs for the '*pulse oximetry done?*' outcome do not overlap with ones from complete case analysis (online supplemental table 4). From the complete case sensitivity analysis, there is a material difference in the estimated association of involvement in any research study variable with the pulse oximetry use at admission outcome (table 3, online supplemental table 4); while the direction of association is the same (positive), the CIs do not overlap. With the age variable, the direction of the association is inverted between the complete case and multiple imputation analyses, and children aged <1 month admitted to the paediatric ward are completely missing in the complete case analysis. These material difference might be attributable to the fact that complete case analysis and our multiple imputation approach make different assumptions of missingness mechanism in the data (ie, missing completely at random mechanism—which is potentially more biased, and a missing at random mechanism,respectively).

‡'*Patient is in any study (Pre-study period)*' reflects the time leading up to any research study starting when no study was being conducted, that is, February 2014–November 2018.

AVPU, Alert, Verbal, Pain, Unresponsive; WAZ, Weight-for-age Z-score.

to 1.13) is positively associated with oxygen prescription at admission (table 3). Oxygen prescription at admission increased with time as part of CIN, having a statistically significant but clinically small month-to-month change (OR: 1.02, 95% CI 1.00 to 1.03). The improvement in the PAR use (OR: 1.23, 95% CI 1.02 to 1.48) also appears to be aiding clinical decision-making in the prescription of oxygen based on the recording of features of respiratory distress.

## DISCUSSION

Over a period of 7 years, with prolonged strikes precluding data collection in 2017, use of pulse oximetry in a population of admissions where 86% have at least one feature indicating risk of hypoxaemia increased slowly with each month of participation in CIN. Most hospitals at the start of their participation in CIN used pulse oximetry in 25% or fewer children, increasing to 81.5% in all hospitals by the final quarter of 2020. Among the admissions prescribed oxygen in 87%, this was not apparently guided by a pulse oximetry value (SpO2) of <90%, the recommended level for intervention to address hypoxaemia in current pneumonia guidelines.

The highly variable patterns of pulse oximetry adoption across hospitals (figure 4) and high ICCs in our models point to the major impact of hospital-level factors on the use of pulse oximetry. These have been previously described and include equipment supply, maintenance, local leadership and clinical efforts to train and promote oximetry use among others.[10] CIN provided regular feedback to hospitals including on pulse oximetry use and SpO$_2$ values throughout a hospital's participation in CIN and engaged subsets of hospitals in research projects that resulted in the placement of one or two additional non-physician clinicians (ie, clinical officers, nurses) in some

hospitals. In the absence of data from other settings it is hard to determine if the increase in pulse oximetry use is part of a wider secular trend (a phenomenon referred to as the 'rising tide'[34]) or associated with CIN participation. Whatever the reasons, it is instructive that adoption of pulse oximetry remains incomplete after 7 years even among inpatient populations with severe illness in moderately sized hospitals. This slow pattern of adoption for an established technology that is widely believed to be beneficial has been observed in high-income settings. It suggests we may only realise the proposed benefits of many technologies including diagnostics much more slowly than is often claimed unless much greater attention is paid to strengthening health service delivery at scale.[35 36]

Overall, approximately 9% of patients were prescribed oxygen at admission, with a slight increase in patients with oxygen prescribed with each month in the CIN. Interestingly, in our models the ICCs exploring oxygen use did not suggest strong hospital-level effects, placing more emphasis on patient-level factors. The number of patients with oxygen prescribed who had no SpO$_2$ recorded is large; more than twice as many children receiving oxygen had pulse oximetry values 90% or more than values <90% (figure 2) although WHO and Kenyan guidelines recommend targeting oxygen use at children with SpO$_2$ <90%.[1] While the source of discrepancy remains largely unknown, it is unlikely that this reflects diagnoses of shock or severe asthma when oxygen use can be recommended when saturations are higher than 90% as these diagnoses are relatively uncommon in Kenyan hospitals.[1 37]

We attempt to integrate our new findings of patient-level factors associated with the use of pulse oximetry and prescription of oxygen at admission with earlier findings that addressed mainly hospital-level factors in figure 5.

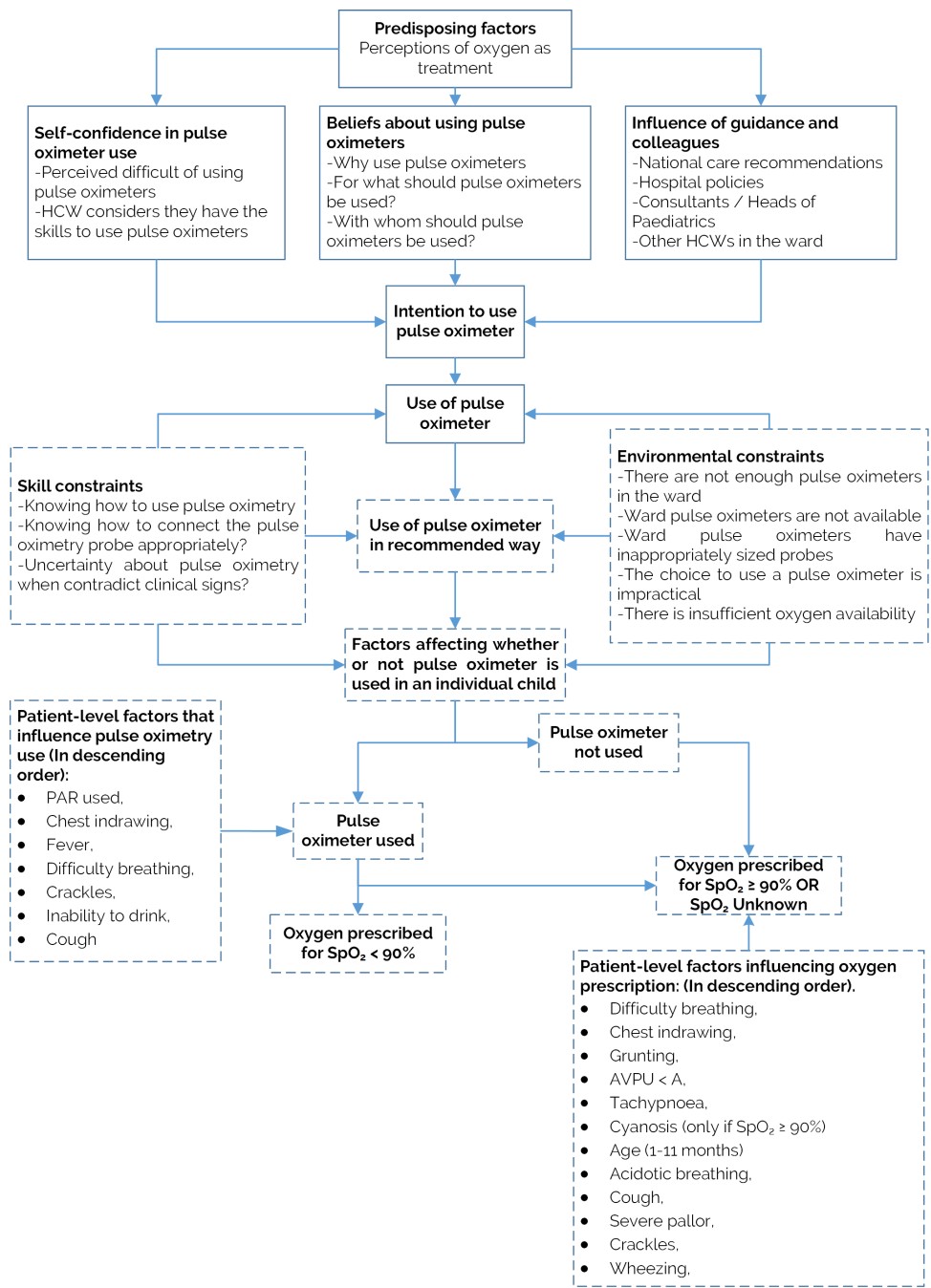

**Figure 5** Adapted integrative model of behavioural prediction (Enoch *et al*[10]). Dashes indicate components of the model that are modified or added to reflect the findings from our new analyses. AVPU, Alert, Verbal, Pain, Unresponsive; HCW: healthcare worker; PAR, paediatric admission record.

Lower chest-wall indrawing is strongly and positively associated with pulse oximetry being done at admission and prescription of oxygen when pulse oximetry is not done. We have previously shown an association between indrawing and mortality for pneumonia admissions and others have recently indicated indrawing may be a useful referral criterion from the community or primary care to a hospital in the absence of pulse oximetry.[5 38] However, randomised trials, and WHO guidance, suggest pneumonia associated with indrawing can be safely treated as an outpatient.[2 37] Our data do not directly inform the debate on the value of indrawing as screening sign

of illness severity but does perhaps suggest it remains an important factor in Kenyan clinicians' decision-making.

Other signs prompting pulse oximeter use are fever, cough, difficulty breathing, crackles and inability to drink (table 3, figure 5). All respiratory signs are associated with oxygen prescription (table 3, online supplemental table 5, figure 5) and a pulse oximetry value <90% was associated with twice the odds of oxygen prescription compared with those whose oxygen saturation was unknown. Where information from pulse oximetry was not available cyanosis, grunting, acidotic breathing, AVPU <A, pallor and tachypnoea appeared to prompt oxygen use (table 3,

figure 5). Several of these signs were not associated with the use of pulse oximetry at admission, perhaps because healthcare providers are taught and then view these signs as indicative of very severe illness (as reported in the literature) requiring intervention with oxygen irrespective of oxygen saturation.[1 37 38] Interestingly, over time, the use of the standardised PAR was associated with twice the odds of a patient having $SpO_2$ measurement and 23% increase in oxygen therapy prescribed at admission, where indicated (table 3). It is possible this effect reflects its role as a prompt to conduct a complete assessment and promote reflection on the use of core interventions recommended in guidelines.

Why infants less than 1 month old have oximetry measured so infrequently remains unknown and warrants more investigation (table 3). It is highly likely that the paediatric wards may not have appropriately sized pulse oximetry probes for the neonates and/or that many of the neonate admissions to the paediatric ward have a diagnosis of sepsis or jaundice which have less clear guidelines on performing routine pulse oximetry on all admissions. Overall, neonates with respiratory problems who mostly tend to be aged closer to time of birth, are sent to newborn wards and thus not meant to be captured in this paediatric admissions dataset (table 2).

It has been well described in many settings that the logical world represented in clinical guidelines is often not replicated in the more complex world of everyday practice.[39–41] While there remain debates in the scientific world around which oxygen saturation threshold for which condition in which setting should guide the use of oxygen,[5 42 43] our data remind us that clinicians are often influenced by many other factors in making such decisions.[44] Clinical decision-making is not straightforward, and it is difficult to capture from routinely collected observational data. Reducing the decision of whether to prescribe or withhold oxygen to a pulse oximetry reading might inadequately reflect the complexities that a clinician encounters at the patient bedside. This has important implications as the presence or prevalence of hypoxaemia, however defined, may be used as proxies for rational use or need for oxygen, respectively. Clinicians' decision to deviate from pulse oximetry defined hypoxaemia thresholds in determining oxygen use, and the subsequent effects this would have on resource use or patient outcomes, remains largely unaddressed in research.[2]

### Strengths and limitations

The data we present are a rare example of long-term tracking of hospital practice in a routine LMIC setting covering common childhood illnesses (online supplemental table 6). The longitudinal nature of our observations offers important insights but also poses challenges. Major changes may occur in the health sector over long periods, as exemplified by the prolonged strikes in Kenya. Many factors ranging from national and local procurement or equipment donations, to changes in local

leadership and hospital's participation in specific studies might influence the adoption of pulse oximetry and use of oxygen we describe. Furthermore, we have no contemporaneous data from Kenyan settings that do not belong to the CIN for comparison and most other studies exploring the use of pulse oximetry are essentially cross-sectional. We do, however, feel our data are at least a useful illustration of the slow and sometimes erratic process of pulse oximetry adoption in LMIC hospitals that has broader implications for thinking on technology adoption.

Another limitation of this study is the absence of new qualitative data especially exploring why oxygen use is often not linked to pulse oximetry values. However, we have previously shown the importance of local clinical leadership in these settings[45] and the continuing challenge of promoting guideline adherence in these complex hospital settings where junior clinicians who manage patients rotate frequently and may err on the side of caution to use interventions rather than risk withholding them.[46–48]

Further challenges of the large-scale, longitudinal nature of our work are data quality and analytic methods. The CIN has made efforts to capture high-quality data through continuous training of clerks linked to error checking procedures and regular engagement with hospitals that include monitoring and feedback.[18 28 29] Nonetheless, imputation procedures that assume a MAR mechanism had to be employed to enable analyses of all eligible case records.[32 49] We also had to employ a complex modelling framework to explore patient-level associations with our outcomes. These attempted to accommodate and account for between hospital variability, within hospital non-linear trends in the outcomes over time and entry into the CIN at different time points. We tested our assumptions where possible (eg, that the strike year was not associated with major changes in practice). However, while our large dataset enabled us to identify many credible associations specific findings should be interpreted cautiously.

### Conclusions

Over time, there has been a progressive increase in the use of pulse oximeters at admission in the Kenyan hospitals we studied. After 7 years, however, use in this high mortality population where almost 9 out of 10 children have least one sign indicating increased risk of hypoxaemia is not universal and major differences exist between hospitals in the adoption of this technology. Guidelines suggest pulse oximetry should determine which children receive oxygen. Our data suggest this is often not the case even in late 2020 during the COVID-19 pandemic, and that many organisational and clinical factors influence whether oximetry is used at all or whether recommended hypoxaemia definitions are used to guide the use of oxygen. As global efforts gain pace to provide pulse oximeters and oxygen systems our data show it is perhaps naïve to expect they will be adopted and employed in the rational ways imagined in guidelines and policies.

**Collaborators** The Clinical Information Network Author Group: The CIN author group who contributed to the network's development, data collection, data management, implementation of audit and feedback and who reviewed and approved this publication include: (1) Central CIN team members: Lynda Isaaka, George Mbevi, Cynthia Khaenzi, Paul Mwaniki, Mercy Chepkirui, John Wainaina, Livingstone Mumelo, Edith Gicheha, Monica Musa, Naomi Muinga and Muthoni Ogola. (2) National partners: Laura Oyiengo, Caroline Mwangi, Fred Were, and (3) Hospital teams: (Paediatricians) Juma Vitalis, Nyumbile Bonface, Roselyne Malangachi, Christine Manyasi, Catherine Mutinda, David Kibiwott Kimutai, Rukia Aden, Caren Emadau, Elizabeth Atieno Jowi, Cecilia Muithya, Charles Nzioki, Supa Tunje, Dr. Penina Musyoka, Wagura Mwangi, Agnes Mithamo, Magdalene Kuria, Esther Njiru, Mwangi Ngina, Penina Mwangi, Rachel Inginia, Melab Musabi, Emma Namulala, Grace Ochieng, Lydia Thuranira, Felicitas Makokha, Josephine Ojigo, Beth Maina, Catherine Mutinda, Mary Waiyego, Bernadette Lusweti, Angeline Ithondeka, Julie Barasa, Meshack Liru, Elizabeth Kibaru, Alice Nkirote Nyaribari, Joyce Akuka, Joyce Wangari; (Nurses) Amilia Ngoda, Aggrey Nzavaye Emenwa, Dolphine Mochache, Patricia Nafula Wesakania, George Lipesa, Jane Mbungu, Marystella Mutenyo, Joyce Mbogho, Joan Baswetty, Ann Jambi, Josephine Aritho, Beatrice Njambi, Felisters Mucheke, Zainab Kioni, Jeniffer, Lucy Kinyua, Margaret Kethi, Alice Oguda, Salome Nashimiyu Situma, Nancy Gachaja, Loise N. Mwangi, Ruth Mwai, irginia Wangari Muruga, Nancy Mburu, Celestine Muteshi, Abigael Bwire, Salome Okisa Muyale, Naomi Situma, Faith Mueni, Hellen Mwaura, Rosemary Mututa, Caroline Lavu, Joyce Oketch, Jane Hore Olum, Orina Nyakina, Faith Njeru, Rebecca Chelimo, Margaret Wanjiku Mwaura, Ann Wambugu, Epharus Njeri Mburu, Linda Awino Tindi, Jane Akumu, Ruth Otieno, Slessor Osok; (HRIOs) Seline Kulubi, Susan Wanjala, Pauline Njeru, Rebbecca Mukami Mbogo, John Ollongo, Samuel Soita, Judith Mirenja, Mary Nguri, Margaret Waweru, Mary Akoth Oruko, Jeska Kuya, Caroline Muthuri, Esther Muthiani, Esther Mwangi, Joseph Nganga, Benjamin Tanui, Alfred Wanjau, Judith Onsongo, Peter Muigai, Arnest Namayi, Elizabeth Kosiom, Dorcas Cherop, Faith Marete, Johannes Simiyu, Collince Danga, Arthur Otieno Oyugi, Fredrick Keya Okoth.

**Contributors** TT: conceptualisation, methodology, software, data curation, visualisation, writing—original draft, writing—reviewing and editing. JA, SA, AA: methodology, investigation, writing—reviewing and editing. GI: writing—reviewing and editing. ME: conceptualisation, methodology, supervision, investigation, writing—reviewing and editing.

**Funding** The work described was supported by multiple funders. These include the Wellcome Trust (#207522) through an award to ME as a Senior Fellowship and a core grant awarded to the KEMRI-Wellcome Trust Research Programme (#092654); by a grant from World Health Organisation: An evaluation in Kenya of the cluster-randomised RTS,S/AS01 malaria vaccine implementation pilot through routine health systems in moderate to high malaria transmission settings (WHO-Project requisition number 2018/854999); by a grant to the NEST program from the John D. and Catherine T. MacArthur Foundation, the Bill & Melinda Gates Foundation, ELMA Philanthropies, and The Children's Investment Fund Foundation UK under agreements to William Marsh Rice University with a sub-agreement to University of Oxford Centre for Tropical Medicine and Global Health; and an award to AA (MR/R006083/1) from the Foreign, Commonwealth and Development Office (FCDO), the MRC, the National Institute for Health Research (NIHR) and Wellcome Trust Joint Global Health Trials Scheme. For the purpose of Open Access, the author has applied a CC-BY public copyright licence to any author accepted manuscript version arising from this submission. The funders had no role in the preparation of this report or the decision to submit for publication.

**Competing interests** None declared.

**Patient consent for publication** Not required.

**Ethics approval** This CIN study was approved by the Kenya Medical Research Institute's (KEMRI) Scientific and Ethics Review Unit (#SERU 3459). Additionally, it was approved by the Ministry of Health, with the Medical Superintendents of participant hospitals given consent for participation. Individual consent for access to de-identified patient data was not required.

**Provenance and peer review** Not commissioned; externally peer reviewed.

**Data availability statement** Data are available upon reasonable request. Data are available upon reasonable request. The source data are owned by the Kenyan Ministry of Health, County Governments and as the data might be used to deidentify hospitals the study authors are not permitted to share the source data directly. Users who wish to reuse the source data can make a request initially through the KEMRI-Wellcome Trust Research Programme data governance committee. This committee will supply contact information for the KEMRI Scientific and Ethical Review unit, County Governments, and individual hospitals as appropriate. The KEMRI-Wellcome Trust Research data governance committee can be contacted on: dgc@kemri-wellcome.org.

**ORCID iDs**
Timothy Tuti http://orcid.org/0000-0002-7915-3004
Jalemba Aluvaala http://orcid.org/0000-0002-0851-3711
Mike English http://orcid.org/0000-0002-7427-0826

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
