## [Reviewer comments · BMJ Open]

ARTICLE DETAILS

TITLE (PROVISIONAL)	Pulse oximetry adoption and oxygen orders at paediatric admission over 7 years in Kenya: A multi-hospital retrospective cohort study
AUTHORS	Tuti, Timothy; Aluvaala, Jalemba; Akech, Samuel; Agweyu, Ambrose; Irimu, Grace; English, Mike

VERSION 1 – REVIEW

REVIEWER	Mukanga, D Bill & Melinda Gates Foundation
REVIEW RETURNED	14-Apr-2021

GENERAL COMMENTS	This paper addresses an important research question, and adds to the scientific body of knowledge. Areas that require clarity or responses: 1. Line 32; what are these organizational issues/weaknesses - provide examples. This also needs to be specific across the paper - staff training, availability of SOPs or Pulse Oximeters, servicing of oximeters?2. In the results section, the authors should tell us what types of pulse oximeters were in use across the facilities as the sensor quality varies, and they do require periodic maintenance. How many oximeters were available to clinicians - this is key before we even talk about consistent use (Nos per hospital, and on the wards in question).3. Some qualitative methods may have been used to provide context around the information presented in this paper.4. Line 350 - the association between hospitals with research program and pulse Ox measurement is good to know, but not useful for policy or action. Why do the authors think they should report on this? Was this a question set a priori and to what end?5. Discussion - the authors can significantly reduce the length of this paper, by avoiding to repeat results in the discussion section. The purpose of this section is to explain results and not to present them a 2nd time e.g. Line 382, no need to present the measure of association and corresponding CIs - rather state the increase month by month and why you think this happened.
--

REVIEWER	Walker, Patrick The University of Melbourne, Dept of Paediatrics
REVIEW RETURNED	14-May-2021

GENERAL COMMENTS	This is an interesting study on an important topic, and presents data that are important for clinicians working with children in LMICs around the world. As such, it merits publication, however
--

	major issues exist which I believe need to be addressed before it can be published. Structural suggestions:  1. The objective described does not fit the body of the paper. The paper is mainly descriptive about the use of oximetry and oxygen, and does not in detail discuss specific factors that influence this (as the objective states). Further, Covid-19 is only peripherally relevant to this topic - I would suggest removing it from the objective, and indeed the paper overall, except perhaps as a contextual comment. 2. Given the point above, if possible I would suggest more detailed exploration of the individual-level, hospital-level, and system-level factors that influence oximetry and oxygen use, as it is clear (as the authors rightly highlight) that they differ considerably from guidelines. If not possible, I would suggest re-framing the objective as noted above. 3. The most interesting result from this study is the low proportion of children that have their SpO2 measured, and the strikingly high number of children who were prescribed oxygen without documented hypoxaemia. This needs to be positioned more centrally in the paper, as the first sentence of the Results section and explored more adequately in the Discussion. 4. It is not clear what the odds ratio of 1.11 means in relation to pulse oximetry use rising each month after hospitals joined the CIN. Does it mean use increased by 11% each month, at a consistent rate? It seems to me that an OR is not an appropriate outcome measure given the independent variable (months in CIN) is continuous. 5. The broader context in which the study took place is not explored in the paper. It would be good to have some information on Kenya's healthcare system, whether oximetry and oxygen are available outside of major hospitals, what referral systems exist for unwell patients and how these are used, and the burden of disease in Kenyan children. 6. Several paragraphs in the Discussion section are not backed up by the Results, and/or are not adequately referenced (see specific comments below). Specific comments: Abstract:  1. Objective does not fit body of paper (see above). Suggest changing to focus on descriptive observation of oximetry and oxygen use, and removing mention of Covid-19. 2. Suggest dot points for outcomes to make it clearer to the reader what the outcomes were. Also need to include what the measures were here. 3. "Association with presenting clinical features" is not an outcome measure - may fit better under the 'Results' sub-heading. 4. Suggest placing the observation that 87% of children who were prescribed oxygen did not have documented hypoxaemia more prominently - this is a very interesting and important finding.
--	--

VERSION 1 – AUTHOR RESPONSE

Reviewer: 1

Dr. D Mukanga, Bill & Melinda Gates Foundation Comments to the Author:

This paper addresses an important research question and adds to the scientific body of knowledge.

Areas that require clarity or responses:

We thank you very much for your comments, your insight has been very informative in making this revised manuscript better. Below are our detailed responses to your comments.

1. Line 32; what are these organizational issues/weaknesses - provide examples. This also needs to be specific across the paper - staff training, availability of SOPs or Pulse Oximeters, servicing of oximeters?

In response to the reviewer's comments, we highlight these organisational issues in both the introduction section where adoption is undermined by inadequate supply and repair of oximeters and if healthcare workers have insufficient training on when, how, and why to use them and interpret their results (lines 69-72), problems also common to oxygen delivery systems. In the discussion section we highlight how adoption is also adversely affected by local leadership and clinical efforts to train and promote oximetry apart from equipment supply and maintenance amongst others (lines 366-369). We also provide reference to qualitative work we had conducted which expounds on these organisational issues [1] and discuss the dissonance between use of oxygen and pulse oximetry values in the discussion (lines 416 – 427). For the purposes of keeping the abstract within the word count limit, we did not itemise the examples in the abstract.

2. In the results section, the authors should tell us what types of pulse oximeters were in use across the facilities as the sensor quality varies, and they do require periodic maintenance. How many oximeters were available to clinicians - this is key before we even talk about consistent use (Nos per hospital, and on the wards in question).

In response to the reviewer's comments, we have updated Supplementary Table 1 to include the counts of the pulse oximeters in use in each of the hospitals. We have provided information on the hospitals where we provided pulse oximeters when they joined CIN in Table 1. Given that this study is set in paediatric wards, our explanation of the available equipment is limited to this ward: there is usually limited information on hospital-wide equipment availability. Also, as updated in Supplementary Table 1, we do not know the brands or specific types, but majority of pulse oximeter machines will be portable devices based on authors' experiences of multiple visits to these facilities over this period.

3. Some qualitative methods may have been used to provide context around the information presented in this paper.

We agree with the reviewer on the benefits of qualitative methods to provide context. The qualitative work which provides more context to the information presented in this paper is referenced throughout the document (lines 69-72, 134-143, 178-183, 317-327, 366-369, 416-430 (Figure 5)). We report the detailed qualitative work elsewhere [1], with this study serves as an extension of the previous work. One limitation of this study is the absence of new qualitative data especially exploring why oxygen use is often not linked to pulse oximetry values. However, we have previously shown the importance of local clinical leadership in these settings [2] and the continuing challenge of promoting guideline adherence in these complex hospital settings where junior clinicians who manage patients rotate frequently and may err on the side of caution to use interventions rather than risk withholding them [3-5]. We have updated the limitation section in line 439-444 to articulate this.

4. Line 350 - the association between hospitals with research program and pulse Ox measurement is good to know, but not useful for policy or action. Why do the authors think they should report on this? Was this a question set a priori and to what end?

In response to the reviewer's comments, for hospitals that were part of research studies, given the additional staff provided and guaranteed availability of working pulse oximeters (Table 1, lines 118-119) this resulted in a contextual difference between hospitals that, for transparency, we felt it was important to report. This also presented an opportunity for examining whether being part of such studies provided material advantage in the uptake and the use of pulse oximetry compared to hospitals whose process of sourcing and maintenance of broken oximeters is reliant on more typical public health system procedures. Our inclusion of the research participation aspect in this report is to make sure hospital level factors which might affect uptake of pulse oximetry use (with a research study that provides oximeters and trained clinical staff to use qualifying as such a factor) are highlighted, and provide context for interpreting the findings (lines 366-369).

Such material differences also help to tease out any added advantage in improving uptake of pulse oximetry from the additional human and material resources and added rigour in clinical practice audit due to hospitals (and by extension individual patients) participating in research trials as highlighted in lines 248-261.

5. Discussion - the authors can significantly reduce the length of this paper, by avoiding repeating results in the discussion section. The purpose this section is to explain results and not to present them a 2nd time e.g. Line 382, no need to present the measure of association and corresponding CIs - rather state the increase month by month and why you think this happened.

We agree with the reviewer and have edited the text in the discussion section to remove repetition of results in terms of reported odds ratio and confidence intervals in lines 361 – 365, 382 - 383.

Reviewer: 2

Dr. Patrick Walker, The University of Melbourne Comments to the Author:

This is an interesting study on an important topic, and presents data that are important for clinicians working with children in LMICs around the world. As such, it merits publication, however major issues exist which I believe need to be addressed before it can be published.

We thank you very much for the thoughtful and encouraging comments you provided. Below are our detailed responses to the changes and clarifications you requested.

Structural suggestions:

1. The objective described does not fit the body of the paper. The paper is mainly descriptive about the use of oximetry and oxygen, and does not in detail discuss specific factors that influence this (as the objective states). Further, Covid-19 is only peripherally relevant to this topic - I would suggest removing it from the objective, and indeed the paper overall, except perhaps as a contextual comment.

In response to the reviewer's comments, we have removed references to COVID-19 in the objectives in the abstract (lines 14-15), and in the introduction (lines 76-79); We have left the remaining three references to COVID-19 in the rest of the paper as contextual comments linked to the use of oxygen and pulse oximetry at the hospital level. We have also reworded our objectives in the abstract (lines 12-15) and in the introduction (lines 76-78) to highlight that our objectives were to characterise adoption and explore specific clinical, patient and hospital factors that might influence pulse oximetry and oxygen use in LMICs over time; We link the findings of this paper to our previous work[1] to highlight useful considerations for entities working on programs to improve access to pulse oximetry and oxygen.

2. Given the point above, if possible I would suggest more detailed exploration of the individual-level, hospital-level, and system-level factors that influence oximetry and oxygen use, as it is clear (as the authors rightly highlight) that they differ considerably from guidelines. If not possible, I would suggest re-framing the objective as noted above.

In response to the reviewer's comments, we have reworded our objectives in both the abstract and the introduction section (lines 12-15, 76-79) to emphasise that the study explored individual- (encompassing clinical and patient level factors in table 2 and 3), and hospital- factors (lines 160-177, Table 1). We also updated lines 127-132 to highlight key system factors relevant to this study such as the promotion of the use of PAR forms at the hospital, the updating of ETAT+ training material nationally to include recommend pulse oximetry as part of sick newborn assessment, and the national dissemination of the updated basic paediatric protocols with updated guidelines on use of pulse oximetry and oxygen use in. The findings in this paper are also linked to previous work on this subject [1] as illustrated by figure 5, and lines 391-397, and 410-415.

3. The most interesting result from this study is the low proportion of children that have their SpO₂ measured, and the strikingly high number of children who were prescribed oxygen without documented hypoxaemia. This needs to be positioned more centrally in the paper, as the first sentence of the Results section and explored more adequately in the Discussion.

In response to the reviewer's comments, we have amended the first paragraph in the discussion to include this key finding (lines 359-365) and amended text in lines 382-390 to discuss why the discrepancy might exist in patients without SpO₂ measure but with oxygen prescription at admission. Together with highlighting this finding in the abstract (lines 25-27) and results lines 279-286, we believe its importance is well captured and that adding this finding at the beginning of the results section while it stems from figure 4 results might compromise the flow of the report. We are cognisant of the challenges of using routine clinical data to expose the sources of this discrepancy in lines 416-427; one of those challenges being how clinical decision-making needs and complexity of bedside patient care might not be adequately reflected by the 'simplified' standard guidelines or captured by routine clinical data collected.

4. It is not clear what the odds ratio of 1.11 means in relation to pulse oximetry use rising each month after hospitals joined the CIN. Does it mean use increased by 11% each month, at a consistent rate? It

seems to me that an OR is not an appropriate outcome measure given the independent variable (months in CIN) is continuous.

We used Odds Ratio because both outcomes were binary (whether pulse oximetry measurement was taken at admission (Yes/No), and whether Oxygen was prescribed at admission (Yes/No)). Our approach to the growth-curve model fitting is within the multilevel modelling framework, with patients nested in hospitals nested in time points (lines 197-202). When the predictor is continuous, like time, it can predict the odds of, for example, having pulse oximetry done at admission as the ratio of those with SpO₂ measurement/ those without SpO₂ measurement per time-point, nested within a hospital. Logistic regression model is a special case of the Generalised Linear Models, and when in log-odds form functions as a linear model capable of modelling associations between the binary outcome and continuous, discrete, and categorical predictors. A primer on modelling such a scenario can be found here: https://link.springer.com/chapter/10.1007/978-3-319-20600-4_37 .

Specifically, for the OR of 1.11, it is correct that there is an 11% increase in the odds that a patient will receive SpO₂ at admission with each additional month in the CIN after adjusting for the other covariates in the model and holding each random effect for an individual patient constant. (Of note we are specific in our use of the term odds ratio to avoid the commonly incorrectly used term 'risk ratio' as this 11% change in odds ratio is not commensurate with an 11% increase in risk ratio per month).

5. The broader context in which the study took place is not explored in the paper. It would be good to have some information on Kenya's healthcare system, whether oximetry and oxygen are available outside of major hospitals, what referral systems exist for unwell patients and how these are used, and the burden of disease in Kenyan children.

We do agree with the reviewer that we have not reported in detail the broader context in which the study took place. We have included lines 93-100 which details how in many LMICs including Kenya, hospital management and monitoring systems are weak, with major human and material resource constraints. These challenges affect hospitals' delivery of inpatient maternal, surgical, and adult medical care as well as paediatric and neonatal care.

These hospitals are essentially all 'first referral level' hospitals predominantly admitting patients that present directly to the facility. A few patients may be formally referred from primary care facilities but there are few functional referral mechanisms such as ambulance systems [6]. Pulse oximetry has generally not been available outside hospitals in the public sector [7]. We have added lines 101-108 to explain this.

Consequently, there is very limited organisational and resource slack to mobilise for any new purpose. Interventions such as oximetry and oxygen seeking to achieve large scale change must therefore either consider how to mobilise new resources or consider what is achievable with limited resources [8, 9]. We also include additional references that described the broader context of the Kenyan healthcare system with reference to the paediatric burden of disease in great detail elsewhere [8, 9] (lines 92-99), including showing pneumonia to be the major killer of children across the country except in settings where malaria is highly endemic (9/18 hospitals in this study) [6] (lines 107-108). They also highlight the inadequate supply and repair of oximeters and if healthcare workers have insufficient training on when, how, and why to use them and interpret their results [1] (lines 69-71).

6. Several paragraphs in the Discussion section are not backed up by the Results, and/or are not adequately referenced (see specific comments below).

Specific comments:

a) Abstract: Objective does not fit body of paper (see above). Suggest changing to focus on descriptive observation of oximetry and oxygen use and removing mention of Covid-19.

In response to the reviewer's comments, we have updated the abstract to remove reference to Covid-19, and in the objective paragraph of the introduction section (lines 12-15, 76-79). We have also reworded the objective to reflect that the focus of the analysis was on the patient, clinical and hospital factors characterising adoption of oximetry and oxygen use over time in hospitals (lines 12-15, 76-79).

b) Suggest dot points for outcomes to make it clearer to the reader what the outcomes were. Also need to include what the measures were here.

In response to the reviewer's comments, we have updated the second dot-point to include the outcomes of this study (lines 43-46). We have also reworded lines 178-180 in the methods section to highlight the two primary outcomes: pulse oximetry use, and, oxygen prescription at admission.

c) "Association with presenting clinical features" is not an outcome measure - may fit better under the 'Results' sub-heading.

In response to the reviewer's comments, we have reworked the lines 19-21, to clarify that the outcomes of interest were pulse oximetry use and oxygen prescription on admission; and that we performed growth-curve modelling to investigate the association of patient factors with study outcomes over time while adjusting for hospital factors.

d) Suggest placing the observation that 87% of children who were prescribed oxygen did not have documented hypoxaemia more prominently - this is a very interesting and important finding.

In response to the reviewer's comments, we have highlighted this finding in the abstract (lines 25-27) and results section in lines 279-286. We have updated the first paragraph in the discussion to include this key finding (lines 359-365) and amended the text in lines 382-390 to discuss this finding.

References

1. Enoch, A.J., et al., *Variability in the use of pulse oximeters with children in Kenyan hospitals: A mixed-methods analysis*. PLoS medicine, 2019. **16**(12): p. e1002987.
2. Nzinga, J., G. McGivern, and M. English, *Examining clinical leadership in Kenyan public hospitals through the distributed leadership lens*. Health policy and planning, 2018. **33**(suppl_2): p. ii27-ii34.
3. Maina, M., et al., *Antibiotic use in Kenyan public hospitals: Prevalence, appropriateness and link to guideline availability*. International Journal of Infectious Diseases, 2020. **99**: p. 10-18.
4. McKnight, J., et al., *Evaluating hospital performance in antibiotic stewardship to guide action at national and local levels in a lower-middle income setting*. Global Health Action, 2019. **12**(sup1): p. 1761657.

5. Ogero, M., et al., *Examining which clinicians provide admission hospital care in a high mortality setting and their adherence to guidelines: an observational study in 13 hospitals*. Archives of disease in childhood, 2020. **105**(7): p. 648-654.
6. Ayieko, P., et al., *Characteristics of admissions and variations in the use of basic investigations, treatments and outcomes in Kenyan hospitals within a new Clinical Information Network*. Archives of disease in childhood, 2016. **101**(3): p. 223-229.
7. Ministry of Health, *Kenya Service Availability and Readiness Assessment Mapping (SARAM)*. 2014, Government of Kenya Nairobi, Kenya.
8. English, M., et al., *Employing learning health system principles to advance research on severe neonatal and paediatric illness in Kenya*. BMJ Global Health, 2021. **6**(3): p. e005300.
9. English, M., et al., *Programme theory and linked intervention strategy for large-scale change to improve hospital care in a low and middle-income country-A Study Pre-Protocol*. Wellcome Open Research, 2020. **5**.

VERSION 2 – REVIEW

REVIEWER	Walker, Patrick The University of Melbourne, Dept of Paediatrics
REVIEW RETURNED	27-Jun-2021

GENERAL COMMENTS	The authors have done an excellent job responding to previous comments. Unfortunately a number of my previous comments do not appear to have been uploaded to the BMJ portal correctly. Nonetheless all structural concerns have been adequately addressed. Some remaining comments:  - Suggest mentioning number of oximeters available at each hospital (e.g. per patient and/or per clinician) in the main body rather than just supplementary material - Low proportion of children being prescribed oxygen based on hypoxia as confirmed with pulse oximetry is still buried deep in the results section. Still suggest making it more prominent as it is the study's most salient finding (aside from factors influencing this, which are quite rightly further down), perhaps as part of an introductory paragraph - Table 3 label (Intercept) is unclear. Suggest editing Table to make it more self-explanatory for readers - Why did infants less than 1mo old have oximetry measured so infrequently? Would be good to explore this as it is a potentially concerning (and interesting) finding - Table 3 - fever/cough/etc all had small effects on pulse oximetry use (despite $p < 0.05$). Worth making this small magnitude of effect clearer in text of Results - it currently reads as though the magnitude of this effect was larger than it actually was - Lines 387-389 - if shock and severe asthma are uncommon, it is unlikely they would have influenced results much. Suggest removing mention of this. It is also interesting that target SpO₂ is >94% in shock - why is this? - Lines 426-428 present a point which is controversial and doesn't really add to the contention of the article. Suggest removing or rephrasing Overall the authors are to be commended. I look forward to seeing this article published.
---

VERSION 2 – AUTHOR RESPONSE

Reviewer: 2

Dr. Patrick Walker, The University of Melbourne Comments to the Author:

The authors have done an excellent job responding to previous comments. Unfortunately, a number of my previous comments do not appear to have been uploaded to the BMJ portal correctly. Nonetheless all structural concerns have been adequately addressed.

We thank you very much for the thoughtful and encouraging comments you provided. Below are our detailed responses to the changes and clarifications you requested.

Structural suggestions:

1. Suggest mentioning number of oximeters available at each hospital (e.g. per patient and/or per clinician) in the main body rather than just supplementary material.

In response to the reviewer's comments, we have added lines 114-115 in the main body to reference that the 18 hospitals included had a median of 1 pulse oximeter(s) per paediatric ward (IQR: 1 - 3) which is also illustrated by Supplementary Table 1.

2. Low proportion of children being prescribed oxygen based on hypoxia as confirmed with pulse oximetry is still buried deep in the results section. Still suggest making it more prominent as it is the study's most salient finding (aside from factors influencing this, which are quite rightly further down), perhaps as part of an introductory paragraph.

In response to the reviewer's comments, we have brought forward this finding to the beginning of the results section, now reflected by lines 223 to 233, including the relevant figure 2.

3. Table 3 label (Intercept) is unclear. Suggest editing Table to make it more self-explanatory for readers.

In response to the reviewer's comments, we have amended the text by adding a footnote in table 3 explaining that the intercept represents the average odds that pulse oximetry is done (or oxygen therapy is prescribed) at admission in this population sample at the first month of being a member of the CIN, when all the predictors are set to their reference levels (e.g., "No").

4. Why did infants less than 1 month old have oximetry measured so infrequently? Would be good to explore this as it is a potentially concerning (and interesting) finding.

Why infants less than one month old have oximetry measured so infrequently remains unknown and warrants more investigation. In lines 416-423 in the discussion section, we speculate that it is highly likely that the paediatric wards may not have appropriately sized pulse oximetry probes for the neonates and/or that many of the neonate admissions to the paediatric ward have a diagnosis of sepsis or jaundice which have less clear guidelines on performing routine pulse oximetry on all admissions. Overall, neonates with respiratory problems who mostly tend to be aged closer to time of birth, are sent to newborn wards and thus not meant to be captured in this paediatric admissions dataset (Table 2).

5. Table 3 - fever/cough/etc all had small effects on pulse oximetry use (despite $p < 0.05$). Worth making this small magnitude of effect clearer in text of Results - it currently reads as though the magnitude of this effect was larger than it actually was.

In line with the reviewer's comment, we have amended the lines that reference these signs and symptoms from Table 3 in lines 316 to 318 in the main body to reflect that patients admitted with signs of fever, cough, difficulty breathing, chest indrawing, crackles, and inability to drink had odds statistically (as opposed to substantively) greater than one of having pulse oximetry measurement taken at admission over time than those without these signs.

6. Lines 387-389 - if shock and severe asthma are uncommon, it is unlikely they would have influenced results much. Suggest removing mention of this. It is also interesting that target SpO₂ is >94% in shock - why is this?

In line with the reviewer's comment, we have reworded the text in lines 384-390 but retained the statement suggesting that that the diagnoses of severe asthma or shock where SpO₂ are relatively uncommon as this is one possible, but unlikely, explanation for the discrepancy in oxygen prescription at admission since a target saturation of $\geq 94\%$ is recommended by WHO guidelines to compensate for the potential of reduced oxygen delivery [1]. Although we note this guidance is not widely disseminated in Kenya. If the reviewer or editors feel this statement should be removed, however, we are happy for this change to be made.

7. Lines 426-428 present a point which is controversial and doesn't really add to the contention of the article. Suggest removing or rephrasing.

In response to the reviewer's comments, we have updated the text in lines 432-434 to explain that clinicians' decision to deviate from pulse oximetry defined hypoxemia thresholds in determining

oxygen use, and the subsequent effects this would have on resource use or patient outcomes, remains largely unaddressed in research.

References

1. Tosif, S. and T. Duke, Evidence to support oxygen guidelines for children with emergency signs in developing countries: A systematic review and physiological and mechanistic analysis. *Journal of tropical pediatrics*, 2017. 63(5): p. 402-413.